# Bio-inspired nano-traps for uranium extraction from seawater and recovery from nuclear waste

Qi Sun[1], Briana Aguila[1], Jason Perman [1], Aleksandr S. Ivanov [2], Vyacheslav S. Bryantsev [2], Lyndsey D. Earl[2], Carter W. Abney[2], Lukasz Wojtas[1] & Shengqian Ma [1]

Nature can efficiently recognize specific ions by exerting second-sphere interactions onto well-folded protein scaffolds. However, a considerable challenge remains to artificially manipulate such affinity, while being cost-effective in managing immense amounts of water samples. Here, we propose an effective approach to regulate uranyl capture performance by creating bio-inspired nano-traps, illustrated by constructing chelating moieties into porous frameworks, where the binding motif's coordinative interaction towards uranyl is enhanced by introducing an assistant group, reminiscent of biological systems. Representatively, the porous framework bearing 2-aminobenzamidoxime is exceptional in sequestering high uranium concentrations with sufficient capacities ($530\,\mathrm{mg\,g^{-1}}$) and trace quantities, including uranium in real seawater ($4.36\,\mathrm{mg\,g^{-1}}$, triple the benchmark). Using a combination of spectroscopic, crystallographic, and theory calculation studies, it is revealed that the amino substituent assists in lowering the charge on uranyl in the complex and serves as a hydrogen bond acceptor, boosting the overall uranyl affinity of amidoxime.

[1] Department of Chemistry, University of South Florida, 4202 E. Fowler Avenue, Tampa, FL 33620, USA. [2] Chemical Sciences Division, Oak Ridge National Laboratory, P.O. Box 2008, Oak Ridge, TN 37831, USA. Correspondence and requests for materials should be addressed to S.M. (email: sqma@usf.edu)

Efficient extraction of uranium has received considerable attention for two imperative reasons: the growing demand for nuclear fuel as well as the complexity of uranium recovery and nuclear waste management[1–7]. Given the uranium deficiency in geological deposits, tremendous efforts have been devoted to extract uranium from seawater, the largest known resource, to safeguard nuclear energy development[8]. Meanwhile, the radioactivity and chemical toxicity of uranium waste poses severe environmental risks, therefore there is a great need to develop remediation technologies in case of incidents or nuclear events[9]. However, uranium is arduous to selectively capture due to its extremely low concentration (several parts per million in nuclear waste water and ~3 parts per billion in seawater (www.world-nuclear.org)) together with the coexistence of a variety of concentrated interfering ions[10]. In addition, the vast volumes of seawater/wastewater management pose additional challenges[1–7]. To accomplish these ambitious tasks, the design of cost-effective and durable adsorbents with a high affinity, fast kinetics, and large capacity towards uranium is crucial[11–18].

Nature has adapted over millennia to recognize specific metal ions with high sensitivity, where well-folded protein scaffolds for chelating ions are assisted by second-sphere interactions[19,20]. Such non-covalent interactions have been shown to play a critical role in tuning the metal-binding affinity and can compensate for modest affinity, realizing strong binding through cooperative interactions. This was clearly demonstrated through the development of engineered proteins, where weakly coordinating species were oriented to form a uranyl binding pocket, achieving femtomolar affinity for uranium, holding promise as potential game-changing technologies in the field of uranyl extraction[19]. However, in view of practical application, there are significant challenges for the viable deployment of engineered proteins such as cost and robustness. Compared with biotechnology, artificial design provides a more deployable format to meet the requirements of real application.

From a chemistry perspective, general principles that are used by nature for realizing these additional mutations to exert control over the binding affinity, include enforcing a correct geometry, exerting charge stabilization, and providing proper hydrogen bond interactions to the distal sites off of the metal centers[21,22]. In this context, it is anticipated that both the stability and selectivity of uranyl chelates will be enhanced when these concepts are included, in addition to coordinative binding with the metal center. Indeed, previous research demonstrates that the affinity of amidoxime functionalized adsorbents for uranyl can be enhanced by incorporation of auxiliary amine or carboxylic acid group, suggesting a potential synergism between these functionalities and the amidoxime group in achieving uranium binding[23,24]. However, the relative position of amidoxime to the auxiliary functionality is undefined in these adsorbents, compromising such a cooperation as well as rational improvement. To maximize potential uranyl binding affinity and selectivity, pre-organization of the complementary array is necessary to be fully achievable for the coordinative binding, with the ideal conformation lower in energy relative to the alternatives.

Taking the abovementioned into account, herein, we demonstrate a promising approach to meet these challenges by creating uranium nano-traps that integrate the metrics of nature and artificial systems with the following features: a bio-inspired uranium coordination environment where the de novo introduced assistant group reinforces the interaction between the chelating site and uranyl, thus enhancing the affinity. Moreover, the binding sites are spatially continued yet highly accessible, which facilitates their cooperation and affords high uranium uptake capacities. In addition to these, they possess a high surface area with hierarchical porosity to enable fast kinetics of uranium adsorption and robustness under various pH environments and high ionic strength solutions, allowing for long-term stable

performance and potential recycling. Such uranium nano-traps can be targeted by constructing judiciously designed bio-inspired chelating systems into porous organic polymers (POPs) due to their exceptional chemical stability as well as flexible molecular design and tunable pore structures[25–30]. These features provide excellent opportunities to introduce hierarchical porosity and offer a high density of the chelating moieties in the resultant adsorbent materials to achieve both efficient binding kinetics and high adsorption capacities to meet the challenges posed by the enormous volumes of wastewater or seawater[2,3].

Given the well-known strong interaction between the amidoxime group and uranyl in conjunction with its cost-effective synthesis[31–33], as well as the excellent hydrogen bonding and electron donating capabilities of the amino group[34,35], they were the coordinative site and reinforcing group of choice, respectively, to demonstrate the proof-of-concept. Considering the importance of spatial distribution of amine and amidoxime for their cooperation, a series of functionalized monomers with an amino substituent in different positions relative to amidoxime were designed to construct into hierarchical porous polymers. Through detailed studies, we showed that the resultant adsorbent constructed with an amino group in the *ortho* position relative to amidoxime displayed extraordinary affinity for uranyl, making it one of the best uranium adsorbents reported thus far. The secondary coordination sphere effects provided by the amino group, including hydrogen bonding interaction and charge stabilization exerted by its electron donating property, account for the observed performance enhancement of the coordinative interaction between amidoxime and uranyl. These findings shed light on the basis for a new and economically competitive strategy for boosting the binding affinity between the adsorbent and metal species for their use in extraction and remediation technology.

## Results

**Synthesis of bio-inspired uranium nano-traps.** In view of materials synthesis for practical applications, free-radical induced polymerization of vinyl-functionalized monomers holds great potential, on account of the monomer tunability together with the adapted and cost-effective synthesis[36,37]. Given that a monomer containing amidoxime (AO) group is not suitable for direct polymerization due to the radical scavenging capability of hydroxylamine, a family of vinyl-functionalized cyano compounds with different amine locations were designed for self-polymerization into highly porous materials followed by post-transformation of the cyano group into amidoxime. Under

**Table 1 Structure of building units and textural parameters of various amidoxime functionalized hierarchical porous polymers**

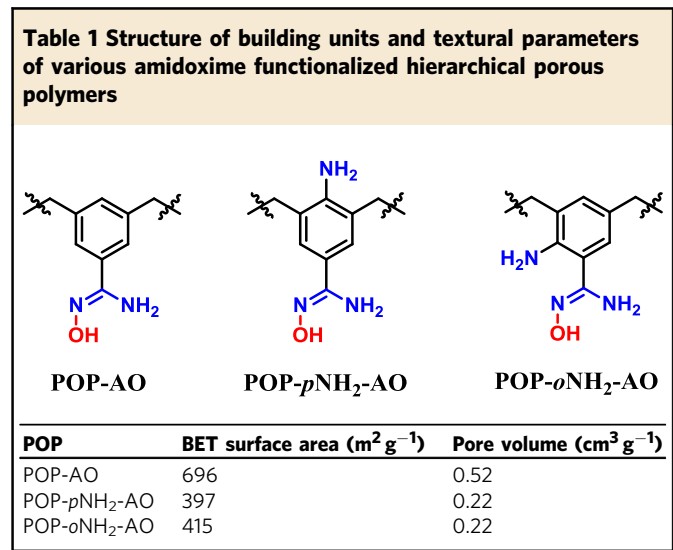

| POP | BET surface area (m² g⁻¹) | Pore volume (cm³ g⁻¹) |
|---|---|---|
| POP-AO | 696 | 0.52 |
| POP-*p*NH₂-AO | 397 | 0.22 |
| POP-*o*NH₂-AO | 415 | 0.22 |

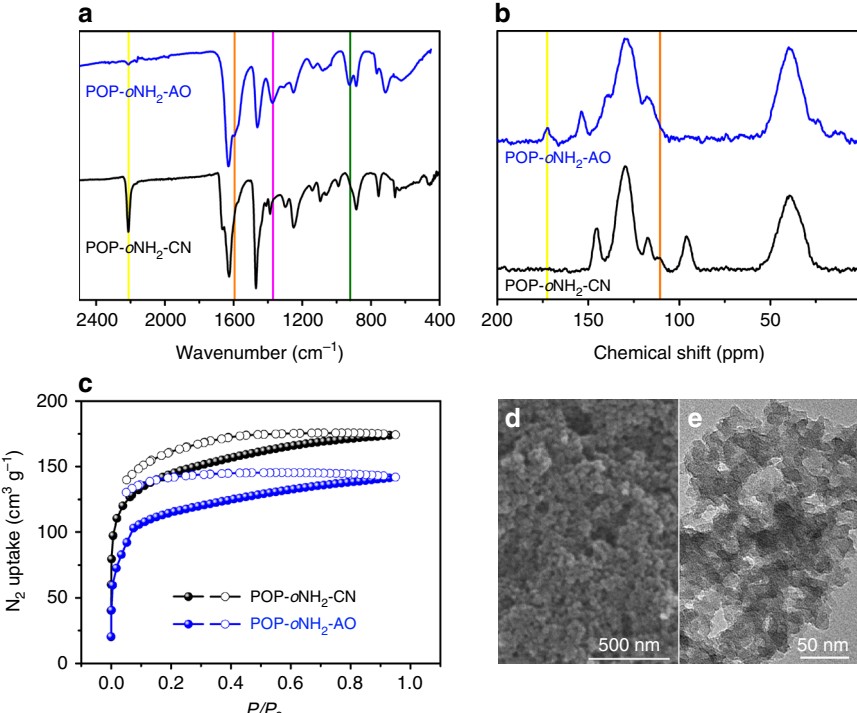

**Fig. 1** Structural characterization. **a** FT-IR spectra, **b** $^{13}$C CP/MAS NMR spectra, **c** N$_2$ sorption isotherms, and **d**, **e** SEM and TEM images for POP-oNH$_2$-AO, respectively

solvothermal conditions in dimethylformamide (DMF) at 100 °C, the polymerization of the monomers in the presence of azobisisobutyronitrile (AIBN), afforded the nitrile-based polymers, which were then amidoximated by treatment with hydroxylamine to afford the AO functionalized polymers (Table 1). It is noteworthy that this process gives rise to the adsorbents in nearly quantitative yields and the vinyl-functionalized cyano compounds can be readily obtained from commercially available reagents in one or two steps, with details provided in the Supplementary Methods. These easy-to-achieve properties give them great promise for practical applications.

**Physiochemical characterization and local structure analysis.** As a representative sample among the synthesized amidoxime-based POPs, the POP bearing 2-amino-benzamidoxime moieties (POP-oNH$_2$-AO) is illustrated thoroughly (See Fig. 1 and the characterization of POP-AO and POP-pNH$_2$-AO are detailed in Supplementary Fig. 1–6). To verify the transformation of nitrile into amidoxime, we carried out Fourier transform infrared spectroscopy (FT-IR) and cross-polarization magnetic-angle spinning (CP/MAS) $^{13}$C nuclear magnetic resonance (NMR) analysis. The disappearance of the nitrile stretch (2206 cm$^{-1}$) coupled with the appearance of C=N (1598 cm$^{-1}$), C–N (1370 cm$^{-1}$), and N–O (917 cm$^{-1}$), characteristic peaks of the amidoxime group, is indicative of the high throughput transformation of nitrile functionality in POP-oNH$_2$-CN to amidoxime (Fig. 1a)[38]. In addition, $^{13}$C CP/MAS NMR spectrum shows the disappearance of the cyano group at 110.7 ppm, replaced by the open-chain amidoxime group at 171.7 ppm (Fig. 1b). To investigate the porosity and pore structures, N$_2$ sorption isotherms were measured at 77 K of the pre-amidoxime and post-amidoxime polymers. As shown in Fig. 1c, both POP-oNH$_2$-CN and POP-oNH$_2$-AO exhibit similar sorption behavior of type I plus type IV, suggesting their hierarchical porous structures comprised of both micropores and mesopores. The adsorption at low pressure ($P/P_0 < 0.01$) is due to the filling of micropores,

while the hysteresis loops at higher relative pressure ($P/P_0 \sim 0.1$ −0.7) is assigned to the presence of mesoporosity in the sample[38,39]. The BET surface areas of POP-oNH$_2$-CN and POP-oNH$_2$-AO were calculated to be 525 and 415 m$^2$ g$^{-1}$, respectively. The hierarchical porosity in these materials can be easily discerned from the scanning electron microscope (SEM) and the transmission electron microscope (TEM) images (Fig. 1d, e). Materials with hierarchical structures are favorable for mass transport, thereby leading to enhanced adsorption kinetics[39,40].

**Uranium sorption studies.** To evaluate the uranium recovery ability, the POPs were initially investigated for the extraction of uranium from the aqueous solutions. Before testing, the adsorbents were treated with a 3% (w/w) aqueous KOH solution at room temperature for 36 h[41]. Equilibrium values were collected by varying the concentrations of uranium in solution at a sorbent/solvent ratio of 0.5 mg mL$^{-1}$ (pH of the solution was optimized at 6, Supplementary Fig. 7) and measuring the subsequent solution concentration after 12 h to guarantee that equilibrium was reached. All three materials exhibited a dramatic color change upon exposure to uranium solutions, turning from white or light brown to orange-red (Supplementary Fig. 8)[42]. The equilibrium adsorption data were well fitted with the Langmuir model, yielding correlation coefficients higher than 0.99 (Fig. 2a and Supplementary Fig. 9). In the uranium concentration range of 36–356 ppm, these materials were determined to have adsorption capacities of 440, 580, and 530 mg of uranium per gram of adsorbent for POP-AO, POP-pNH$_2$-AO, and POP-oNH$_2$-AO, respectively (The explanation of POP-pNH$_2$-AO affording the highest uranium uptake capacity from the aqueous solutions, please see Supplementary Note 1). It is worth noting that, in addition to high adsorption capacities, these POP-based sorbents also possessed extremely rapid capture capabilities, as evidenced by all three adsorbents reaching greater than 90% of their equilibrium capacity after 1 h of contact time (Fig. 2b). Particularly, in the case of POP-oNH$_2$-AO, approximately 95% of the equilibrium

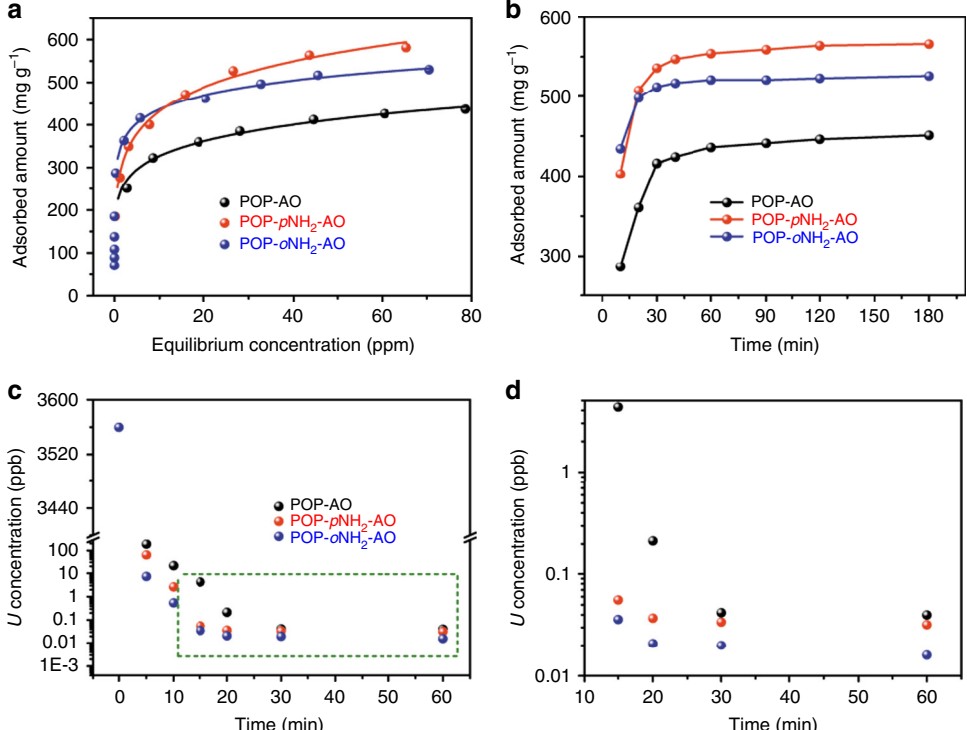

**Fig. 2** Uranium adsorption isotherms and kinetics investigations. **a** Uranium sorption isotherms for POP-based adsorbents. The lines are fit with the Langmuir model; all the fits have $R^2$ values higher than 0.99. **b** The kinetics of uranium adsorption from aqueous solution with an initial concentration (7.56 ppm, 400 mL), at pH ~6, and adsorbent material (3 mg). **c** Uranium removal kinetics with an initial concentration of 3560 ppb (pH ~6) at a $V/m$ ratio of 50,000 mL g$^{-1}$. **d** Enlarged section of green rectangle in **c**

capacity can be achieved within 20 min. This is in stark contrast to the lengthy contact times required for other adsorbents, which routinely range from several hours to as much as several days[43]. We attribute the high adsorption capacities and rapid kinetics to the synergistic effects arising from densely populated chelating groups coupled with hierarchical pores allowing for rapid diffusion of uranyl throughout the materials. More significantly, due to the robustness and chemical stability of the adsorbents (Supplementary Fig. 10), they can be easily regenerated by treating the uranyl-laden materials with a $Na_2CO_3$ (1 M) solution. For example, the adsorption performance of POP-$o$NH$_2$-AO was maintained for at least two cycles, giving rise to uranium uptake capacities as high as 520 and 530 mg g$^{-1}$, respectively.

Considering the observed discrepancy in uptake capacity, whereby POP-AO with higher surface area in comparison with POP-$p$NH$_2$-AO and POP-$o$NH$_2$-AO does not show superior adsorption performance, we therefore reasoned that rather than the textural features, the different coordination environment of these materials seem to be responsible for such differences. Moreover, by contrast with the uranium adsorption isotherms and kinetics of POP-AO, much steeper adsorption profiles and faster kinetics are afforded by POP-$p$NH$_2$-AO and POP-$o$NH$_2$-AO, which are suggestive of their stronger affinity towards uranyl, thereby facilitating the adsorption occurrence. Together, given the superior performance of POP-$o$NH$_2$-AO and POP-$p$NH$_2$-AO to that of POP-AO, it is thus indicated that the introduction of the amino group can enhance the sorption performance of amidoxime-based sorbents.

In this context, to study the contribution of the amino group, tests were focused on the uranyl-sequestration efficiency of these adsorbents along with the kinetics of this process (Fig. 2c, d). Remarkably, more than 99.99% of the uranium was removed within 10 min after treating uranium-spiked water (3560 ppb at pH ~6.0) with POP-$o$NH$_2$-AO and POP-$p$NH$_2$-AO. At this time,

the remaining uranium concentration after treatment with POP-AO was about two orders of magnitude higher than that of the amine-contained sorbents and there was still a large disparity in concentration after reaching equilibrium, underscoring the benefit of the introduction of the amino functionality to enhance the uranyl adsorption performance. It is noteworthy that POP-$o$NH$_2$-AO clearly outperforms POP-$p$NH$_2$-AO with regard to both kinetics and removal efficiency. Taking into account that they have similar surface areas, as well as density and chemical functionality, the disparity in adsorption performance should be ascribed to the difference from the relative position of the amino group to the amidoxime.

To confirm the superior performance of the adsorbent with an amino group in the vicinity of amidoxime, we decided to compare their capability for remediation of real world water samples. Given the vital importance of drinking water safety, we therefore took potable water, which was intentionally contaminated by traces of uranium (1000 ppb) as an example. As shown in Supplementary Fig. 11, the decontamination of uranium can be rapidly accomplished by POP-$o$NH$_2$-AO and the residual uranium concentration decreased to as low as 2 ppb within 5 min [pH ~7, volume ($V$) of solution to mass ($m$) of adsorbent ($V$: $m$ = 50,000 mL g$^{-1}$)], well-below the acceptable limit of 30 ppb defined by the US Environmental Protection Agency (EPA) for potable water[18]. In contrast, nearly two and four times longer duration time are required in order to reduce uranium concentrations to the same level using POP-$p$NH$_2$-AO and POP-AO, respectively. This improved performance emphasizes the significance of the amino group and its position towards uranyl extraction. Impressively, increasing the $V$:$m$ ratio to 500,000 mL g$^{-1}$, POP-$o$NH$_2$-AO still was able to reduce the uranium of the aforementioned contaminated water to less than 1 ppb within 3 h (Supplementary Fig. 12).

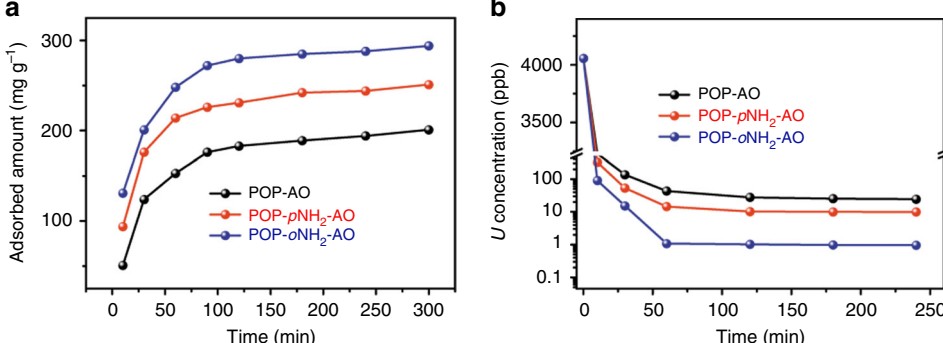

**Fig. 3** Uranium sorption from simulated seawater. **a** The kinetics of uranium adsorption for various adsorbents in simulated seawater solutions with an initial uranium concentration of 10.3 ppm (400 mL) and adsorbent material (3 mg). **b** The kinetics of uranium removal efficiency from simulated seawater spiked with uranyl (4056 ppb) at $V{:}m = 2000\ \mathrm{mL\ g^{-1}}$

With this success, we next evaluated the ability of POP-$o$NH$_2$-AO under more complex conditions to eliminate uranium in the presence of various interfering ions, such as transition heavy metal ions (Cu$^{2+}$, Fe$^{3+}$, Co$^{2+}$, Pb$^{2+}$, Zn$^{2+}$), lanthanides (La$^{3+}$, Ce$^{3+}$, Sm$^{3+}$), radioactive ions (Cs$^+$, Sr$^{2+}$), as well as common ions (Mg$^{2+}$, Ca$^{2+}$). The adsorption tests were performed using a potable water sample containing uranium and the ions aforementioned with nearly equal concentrations (1000 ppb). It is shown that after a single treatment, uranium was removed with over 99% efficiency by POP-$o$NH$_2$-AO when added at a phase ratio of 40,000 mL g$^{-1}$ over 1 h contact time. Impressively, even in the presence of a large excess of aforementioned interfering ions (500 times over uranium, ca. 500 ppm), POP-$o$NH$_2$-AO still can reduce the uranium concentration from 1 ppm to less than 10 ppb at a phase ratio of 10,000 mL g$^{-1}$ within 1 h, indicative of its high selectivity towards uranium. The efficient detoxification of radioactive uranium presented, herein, is of great importance and this study may lead to the development of techniques to treat real water samples suffering from uranium poisoning.

**Uranium capture from simulated seawater and seawater.** Given that nuclear power will be crucial for future low carbon energy generation, there is a strong motivation to develop adsorbent materials that efficiently seize uranium stocks from seawater, a sustainable alternative to traditional mining practices for nuclear fuel ores. In this regard, to demonstrate their prospective recovery of uranium from seawater, we initially applied this series of polymers to test uranium sequestration from a solution spiked with ~10.3 ppm of uranium in the presence of excess Na$^+$ and HCO$_3^-$ (NaCl 25.6 g L$^{-1}$ and NaHCO$_3$ 0.198 g L$^{-1}$) to simulate seawater. Experiments found that POP-$o$NH$_2$-AO exhibited an exceptional ability to adsorb uranium species rapidly with a capacity up to 290 mg g$^{-1}$; whereby equilibrium was reached within 300 min (Fig. 3a). Moreover, POP-$o$NH$_2$-AO can efficiently decrease the uranium concentration to an extremely low level (ca. 1 ppb, removal capacities ~99.9%, $V{:}m = 2000\ \mathrm{mL\ g^{-1}}$, Fig. 3b), which is promising as a uranium adsorbent material for applications in enriching naturally occurring uranium in seawater. Under identical conditions, POP-$p$NH$_2$-AO and POP-AO give rise to adsorption capacities of 250 and 200 mg g$^{-1}$, respectively. In addition to the relatively lower adsorption amount in comparison with POP-$o$NH$_2$-AO, they also exhibit inferior removal efficiencies, supported by the higher residual uranium concentrations of around 10 and 25 ppb, respectively. The distribution coefficient value ($K_d$) is useful for evaluating the affinity of an adsorbent to metal species under specific conditions; the $K_d$ values of these materials were thus calculated with 4056 ppb uranium in simulated seawater solution (10 mL) in the

presence of adsorbents (5 mg). Impressively, POP-$o$NH$_2$-AO demonstrated the highest $K_d$ value of $8.36 \times 10^6$ mL g$^{-1}$ among the samples tested in this work (calculated after reaching equilibrium), which is an order of magnitude higher than the other two materials (POP-$p$NH$_2$-AO, $K_d = 8.18 \times 10^5$ mL g$^{-1}$ and POP-AO, $K_d = 3.28 \times 10^5$ mL g$^{-1}$). As calculated by Eq. 1 below:

$$K_d = \left(\frac{C_i - C_e}{C_e}\right) \times \frac{V}{m},\qquad(1)$$

where $V$ is the volume of the treated solution (mL), $m$ is the amount of adsorbent (g), $C_i$ is the initial concentration of uranium, and $C_e$ is the equilibrium concentration of uranium.

After demonstrating the ability of those adsorbents to extract uranium from simulated seawater, we examined their performance in real seawater samples. This is an ambitious task as seawater is characterized by substantially high ionic strength, including a variety of interfering ions, and extremely low uranium concentration (approximately $3.3\ \mu\mathrm{g\ L^{-1}}$). Initial studies were performed with seawater samples spiked with 10.3 ppm uranium. Reduced uptake capacities from 20 to 30% for POP-AO and POP-$p$NH$_2$-AO were detected relative to the values obtained from the uranium spiked simulated seawater, affording 143 and 202 mg uranium per gram of adsorbent, respectively. By constrast, less than 5% decrease in uptake capacity was observed (290 vs. 276 mg g$^{-1}$) for POP-$o$NH$_2$-AO, furthering confirming its excellent affinity towards uranium.

Encouraged by the aforementioned results, we next sought to determine the adsorption ability of naturally occurring UO$_2^{2+}$ in seawater. Five milligrams of adsorbent was immersed separately in a tank containing 5 gallons of seawater and shaken at room temperature. After 56 days of seawater exposure the amount of uranium enriched in the adsorbent was determined by ICP-MS analysis after being digested by aqua regia. POP-$o$NH$_2$-AO, POP-$p$NH$_2$-AO, and POP-AO demonstrate uranium uptakes of 4.36, 2.27, and 1.32 mg per gram of adsorbent, respectively, confirming the superior performance of POP-$o$NH$_2$-AO and thereby showing great promise for practical applications (comparison with the representative reported results in uranium capture are listed in Supplementary Table 1). Worthy of note, the uptake capacity of POP-$o$NH$_2$-AO is approximately three times higher than the reported value for a benchmark adsorbent reported by Japanese scientists[44]. The results obtained indicate a remarkable efficiency for POP-$o$NH$_2$-AO to extract this extremely low level of uranium and are encouraging for its use for mining nuclear fuel from seawater.

**Crystallographic and density functional theory calculation studies.** Improvements towards uranium adsorption with an amidoxime-based sorbent are evident with the addition of an amino

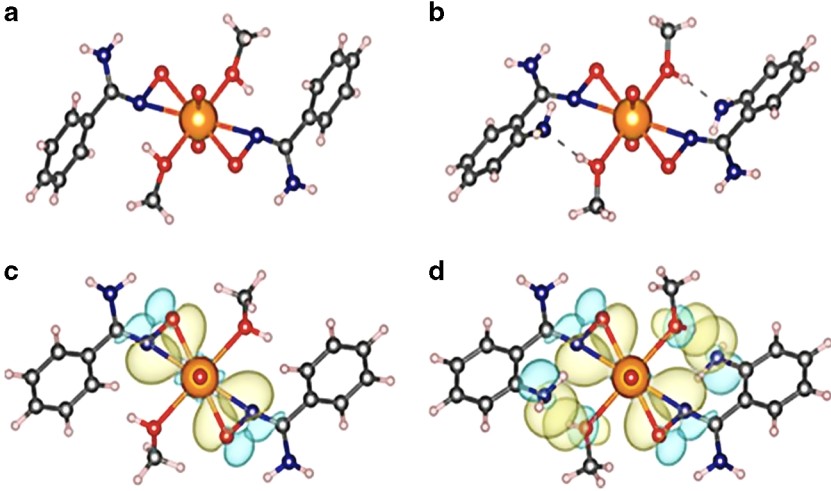

**Fig. 4** Crystal structures of uranyl complexes. **a, b** Single crystal structures of $UO_2(AO)_2(MeOH)_2$ and $UO_2(oNH_2\text{-}AO)_2(MeOH)_2$, respectively. **c** DFT optimized structure of $UO_2(AO)_2(MeOH)_2$ with dative U–O and U–N σ-bonds in $UO_2(AO)_2(MeOH)_2$. **d** DFT optimized structure of $UO_2(oNH_2\text{-}AO)_2(MeOH)_2$ with dative U–O and U–N σ-bonds along with second sphere hydrogen bonding interactions, characterized by overlap of the *p*-type amino lone pair and the methanol O–H σ* orbital

group together with its relative location, as it showed a tenfold improvement from *para* to *ortho* substitution in terms of $K_d$ value. To discern these phenomena, we investigated structure–property relationships of functional groups of the POP-AO, POP-*p*NH$_2$-AO, and POP-*o*NH$_2$-AO adsorbents. In this regard, small molecular ligands, benzamidoxime (AO), 4-aminobenzamidoxime (*p*NH$_2$-AO), and 2-aminobenzamidoxime (*o*NH$_2$-AO) were synthesized and tested for the complexation with uranyl. Single crystals of $UO_2(AO)_2(MeOH)_2$ (Fig. 4a) and $UO_2(oNH_2\text{-}AO)_2(MeOH)_2$ (Fig. 4b and Supplementary Fig. 13) were readily obtained by slowly evaporating a 1 mL methanol solution containing dissolved AO/*o*NH$_2$-AO and $UO_2(NO_3)_2$·6H$_2$O (2:1 molar ratio) with 50 μL trimethylamine, however, any attempts to prepare $UO_2(pNH_2\text{-}AO)_2(MeOH)_2$ suitable for X-ray crystallographic studies were unsuccessful. As seen from Fig. 4a, b, X-ray structures of the $UO_2(AO)_2(MeOH)_2$ and $UO_2(oNH_2\text{-}AO)_2(MeOH)_2$ complexes reveal $\eta^2$ (O, N) binding of the amidoxime moiety to uranyl, which is consistent with the coordination motif observed in DFT calculations (Fig. 4c, d) and previously reported uranyl-amidoxime species[45]. The corresponding U–O and U–N bond lengths in $UO_2(oNH_2\text{-}AO)_2(MeOH)_2$ were found to be 0.01 and 0.02 Å shorter than those of $UO_2(AO)_2(MeOH)_2$ (Supplementary Tables 2 and 3), confirming the stronger bonding between uranyl and the *o*NH$_2$-AO ligand. Furthermore, the $UO_2(oNH_2\text{-}AO)_2(MeOH)_2$ complex is additionally stabilized by relatively short (2.665 Å) hydrogen bonds between the amino groups of *o*NH$_2$-AO and solvent molecules. DFT calculations, Fig. 4d, confirm that this interaction persists in the absence of packing forcers and long-range electrostatic effects (see Supplementary Fig. 13–15 and Supplementary Tables 4 and 5 for additional details). We were also able to find a complex not involved in the hydrogen-bonding interaction with the amino groups, but this was 4.7 kcal mol$^{-1}$ less stable. Natural bond orbital (NBO) analysis clearly identifies only conventional ligand-$UO_2^{2+}$ dative σ-bonds in $UO_2(AO)_2(MeOH)_2$ (Fig. 4c), yielding the second-order stabilization energies ($E^{(2)}$) of 176–278 kcal mol$^{-1}$. In addition to coordinative binding, the uranyl complex with *o*NH$_2$-AO ligands also exhibits strong second-sphere hydrogen bonding interactions ($E^{(2)}$ = 40.0 kcal mol$^{-1}$), as exemplified by the overlap between nitrogen *p*-type lone pair of the amino group and the σ* orbital of the methanol O–H bond (Fig. 4d). This phenomenon of hydrogen bond stabilization is reminiscent of that seen in biological systems and protein

receptors[46]. Although the stability gain contributed by hydrogen-bonding interactions may not be particularly large, especially when compared with the magnitude of the metal binding interactions, these secondary interactions, nevertheless, can play a decisive role in achieving stronger coordination of *o*NH$_2$-AO with uranyl compared to AO and *p*NH$_2$-AO.

The analysis of bond distances and orbital interactions in the representative uranyl crystal and DFT optimized structures provides useful metrics for rationalizing the difference in adsorption behavior of AO, *p*NH$_2$-AO, and *o*NH$_2$-AO-based polymers. Thermodynamic analysis of complexation in aqueous environments can provide a further step towards understanding the differences in the performance of the studied polymers. To this end, the key thermodynamic parameters, such as p$K_a$ of the ligands and stability constants (log *β*) of the respective uranyl complexes, were computationally obtained through our recently developed protocols that achieve high accuracy in predicting aqueous p$K_a$ (root-mean-square deviation from experiment (RMSE) < 0.5 p$K_a$ units)[47] and log *β* (RMSE < 1.0 log units)[48] (see the Methods section and Supplementary Fig. 16 and 17). The results confirm that the highest stability is attained in the uranyl complex with the *o*NH$_2$-AO ligand (log *β*$_2$ = 22.55). Since the electron-donating effect of the amino group at the *ortho* position of the aromatic ring is weaker than that at the *para* position, a slightly lower stability of the complex with the *p*NH$_2$-AO ligands (log *β*$_2$ = 22.47) is a direct consequence of the absence of the second-sphere inter ligand hydrogen-bonding interactions that are only manifested in the complex with *o*NH$_2$-AO (Supplementary Fig. 18). The predicted p$K_a$ values are particularly revealing in rationalizing the complexation properties of the studied ligands. We find that *o*NH$_2$-AO is the least basic ligand, with p$K_a$ = 11.31 that is >1 log unit lower than in AO (p$K_a$ = 12.37) and *p*NH$_2$-AO (p$K_a$ = 12.49). This difference in the basicity can be attributed to the presence of a strong intramolecular hydrogen bond between the amino group at the *ortho* position and amidoximate, which significantly stabilizes the anionic form of *o*NH$_2$-AO (Supplementary Fig. 18). Similar effect was observed in salicylaldoxime, where the decreased basicity of the phenolate group was attributed to hydrogen-bonding interaction with the aldoxime group[48]. Since the complexation of amidoxime with uranyl takes place by displacing a proton[49], the low p$K_a$, can be considered as another advantage of *o*NH$_2$-AO over *p*NH$_2$-AO and AO.

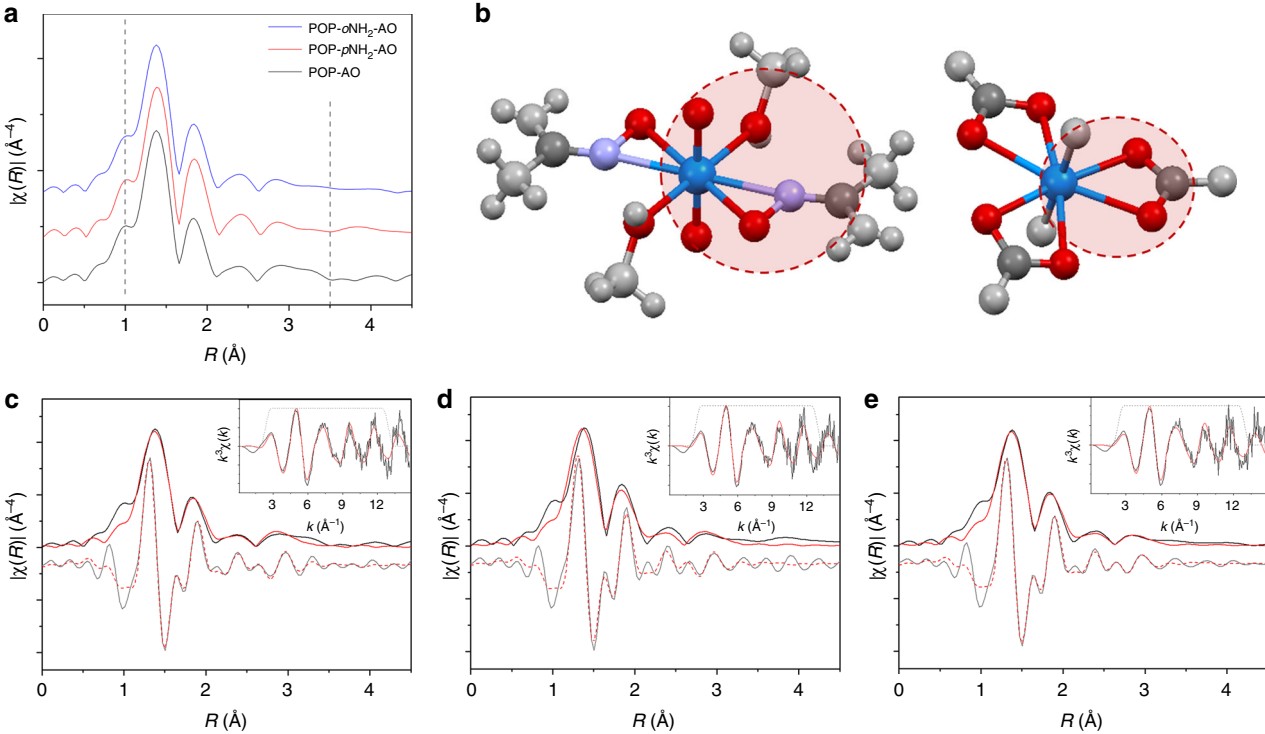

**Fig. 5** EXAFS data, fits, and scattering paths. **a** A comparison of EXAFS spectra for U@POP-AO, U@POP-pNH₂-AO, and U@POP-oNH₂-AO revealing significant qualitative similarities. **b** Graphical depictions of the crystal structures used to generate theoretical scattering paths to fit the EXAFS data. The dashed red circle displays the coordination shells for amidoxime- and carbonate-bound uranyl. Atoms displayed in light gray are not used to generate scattering paths but are included to display the complete molecular structure. EXAFS data and fits for **c** U@POP-AO, **d** U@POP-pNH₂-AO, and **e** U@POP-oNH₂-AO. The graphical depiction of the amidoxime species is displayed by each fit for clarity. The magnitude of the Fourier transform is plotted in black, the real space component in gray, and fits in solid and dashed red, respectively. In all plots the inset is the $k^3$-weighted $\chi(k)$ data and corresponding fit

In order to compare the effectiveness of AO, $p$NH₂-AO, and $o$NH₂-AO to sequester uranyl in the presence of competing ions, species distribution diagrams (Supplementary Fig. 19) were generated by incorporating ligands' p$K_a$ values and the stability constants with uranyl. For amidoxime ligands equilibrium constants were calculated in the present work, while for carbonate and hydroxides they were taken from the literature (Supplementary Table 6)[50,51]. The uranyl and carbonate concentrations were kept at $4.38 \times 10^{-5}$ M and 0.0023 M, respectively, to reproduce the composition of the seawater simulant used in our experiments (Fig. 3). At these conditions and pH ~7–8, the dominant species in the solution are the $UO_2(CO_3)_2^{2-}$ and $UO_2(CO_3)_3^{4-}$ species, and it is with the $CO_3^{2-}$ anion that the AO, $p$NH₂-AO, and $o$NH₂-AO ligands must compete to bind the $UO_2^{2+}$ cation. The speciation diagrams (Supplementary Fig. 19) show that in the presence of 0.001 M ligand concentration, 100% $UO_2^{2+}$ is complexed by $o$NH₂-AO, while $p$NH₂-AO and AO are able to displace only ~75 and ~50% $UO_2^{2+}$, respectively, from the uranyl tricarbonate complex, which is qualitatively consistent with the experimental observations in simulated seawater discussed above. Therefore, the computational results clarify that the superior uranium extraction performance of the POP-$o$NH₂-AO sorbent is due to the synergistic effects between the electron donating and hydrogen bonding capability of the amino group of the $o$NH₂-AO functionality, promoting stronger complexation with $UO_2^{2+}$.

**Investigation of uranyl binding interactions in the adsorbents.** The presented single-crystal X-ray and computational data provide great confidence in the metal binding behavior of AO, $p$NH₂-AO, and $o$NH₂-AO small molecules; however, polymer morphology might have a profound effect on the established

binding modes. Thus, direct validation of the U-binding environment on representative adsorbents is indispensable for achieving definitive conclusion. The materials were further investigated for their complexing behavior towards uranyl. To detect uranium inclusion within these adsorbents, XPS spectroscopy and elemental distribution mapping attached by SEM were performed (Supplementary Figs. 20–22). The appearance of U 4f signals in these reacted samples (U@POP-AO, U@POP-$p$NH₂-AO, and U@POP-$o$NH₂-AO) verified the existence of uranium species. Elemental distribution mapping results showed a homogeneous distribution of captured uranium species throughout each sample. To examine the chemical binding of uranyl in these adsorbents, IR spectroscopy was carried out. In the IR spectra, the new bands at ~903 cm⁻¹ for POP-AO and POP-$p$NH₂-AO as well as 912 cm⁻¹ for POP-$o$NH₂-AO in the uranium included samples are assigned to the antisymmetric vibration of $[O=U=O]^{2+}$ (Supplementary Fig. 23). These peaks have a significant red-shift compared to the corresponding peak in $UO_2(NO_3)_2\cdot6H_2O$ (~960 cm⁻¹, Supplementary Fig. 24)[52], indicating that strong interactions exist between uranyl and the functional groups in these polymers.

To gain further insight, we applied X-ray absorption fine structure (XAFS) spectroscopy to investigate the uranyl binding environment in these three amidoxime-functionalized POPs. Analysis of the extended XAFS (EXAFS) spectra, as displayed in Fig. 5a, reveal that all POPs bind uranyl in a similar fashion. Fits of the EXAFS spectra were achieved by calculation of theoretical scattering paths with FEFF 6 using structure models obtained from the small molecules displayed in Fig. 5b. Direct scattering paths were considered for all atoms in the uranyl first and second coordination sphere, while multiple scattering paths from the axial U=O were also included. A bottom-up approach was

utilized to fit the data, where the coordination number ($N$), change in scattering half-path length ($\Delta r$), and mean squared relative displacement ($\sigma^2$) were local parameters, while amplitude reduction factor ($S_0^2$) and change in absorption edge ($\Delta E_0$) were global parameters to all scattering paths. A detailed discussion of scattering paths and fits are provided in the Supplementary Methods.

As revealed by the refined fits of all three data sets, consistent bond lengths and coordination numbers were obtained for each sample, confirming the X-ray determined and computationally predicted $\eta^2$-binding mode for all adsorbents[53,54]. Furthermore, analysis of the refined coordination numbers support a local uranyl environment containing two amidoxime ligands and one carbonate, largely supporting a 2:1 binding mode (Supplementary Table 7), which reinforces the reasonability of DFT calculation.

## Discussion

In summary, we have developed a promising strategy for the synthesis of highly efficient uranium adsorbent materials. To improve the uptake capacity of the porous adsorbents, the monomer units were designed to allow the synthesis of a high surface area material while conserving a sufficiently high density of chelating groups. To enhance the affinity of chelating groups to uranyl ions, an amino group was de novo introduced to improve coordinative binding interactions by providing a secondary coordination sphere, which alters the electron density of the complex to lower the overall charge on uranyl and provides an additional site allowing a hydrogen bond to align uranyl species in a favorable coordination mode. Adsorption results showed that the porous adsorbent synthesized using 2-aminobenzamidoxime, yielding POP-$o$NH$_2$-AO, acted as a highly efficient uranyl scavenger that was effective in capturing uranium from polluted water and for selectively extracting it from seawater. The underlying principles contributed to its superior performance were revealed by collaborative experiments including spectroscopic, crystallographic, and DFT calculation studies. This proof-of-concept study is important because it affords an amenable route to bridge natural and artificial systems. Moreover, this strategy is practically feasible and thus provides a new direction towards the development of efficient, economic, and applicable adsorbents for uranium capture. Studies aimed at extending this strategy to incorporate other types of reinforcement groups into the coordinative functionalities of the adsorbent materials for metal species sequestration applications are currently underway in our laboratory to further understand the synergistic adsorption effect.

## Methods

**Materials and measurements**. Commercially available reagents were purchased in high purity and used without purification. $^1$H NMR spectra were recorded on a Bruker Avance-400 (400 MHz) spectrometer. Chemical shifts are expressed in ppm downfield from TMS at $\delta = 0$ ppm, and $J$ values are given in Hz. $^{13}$C (100.5 MHz) cross-polarization magic-angle spinning (CP-MAS) NMR experiments were recorded on a Varian infinity plus 400 spectrometer equipped with a magic-angle spin probe in a 4-mm ZrO$_2$ rotor. Nitrogen sorption isotherms at the temperature of liquid nitrogen were measured using Micromeritics ASAP 2020 M and Tristar system. The samples were outgassed for 1000 min at 80 °C before the measurements. Scanning electron microscopy (SEM) and energy dispersive X-ray spectroscopy (EDX) mapping were performed on a Hitachi SU 8000. Transmission electron microscope (TEM) image was performed using a Hitachi HT-7700 or JEM-2100F field emission electron microscope (JEOL, Japan) with an acceleration voltage of 110 kV. XPS spectra were performed on a Thermo ESCALAB 250 with Al Kα irradiation at $\theta = 90°$ for X-ray sources, and the binding energies were calibrated using the C1s peak at 284.9 eV. IR spectra were recorded on a Nicolet Impact 410 FTIR spectrometer. ICP-OES was performed on a Perkin-Elmer Elan DRC II Quadrupole. ICP-MS was performed on a Perkin-Elmer Elan DRC II Quadrupole Inductively Coupled Plasma Mass Spectrometer. Details of X-ray Absorption Fine Structure (XAFS) Spectroscopy, X-ray Crystallography, and Computational studies are given in the Supplementary Methods.

**Synthesis of nitrile-based porous organic polymers**. In a typical run, 3,5-divinylbenzonitrile was dissolved in DMF (10 mL), followed by the addition of free radical initiator AIBN (0.025 g). The mixture was transferred into 20 mL autoclave and maintained for 24 h at 100 °C. A white solid product (quantitative yield) was obtained after being washed with ethanol and dried under vacuum at 50 °C for 24 h, which was denoted as POP-CN. The synthetic procedures of POP-$p$NH$_2$-CN and POP-$o$NH$_2$-CN are similar to those of POP-CN, except that 4-amino-3,5-divinylbenzonitrile or 2-amino-3,5-divinylbenzonitrile (1.0 g) was used instead of 3,5-divinylbenzonitrile. Note: the purity of the synthesized monomers were confirmed by liquid NMR analysis (Supplementary Fig. 25).

**Synthesis of amidoxime functionalized porous polymers**. As a typical synthesis recipe, 0.2 g of POP-CN was swollen in 20 mL of ethanol for 10 min, followed by the addition of 0.5 g of NH$_2$OH·HCl and 0.75 g of N(CH$_2$CH$_3$)$_3$. After being stirred at 70 °C for 48 h to convert the nitrile into amidoxime, the mixture was filtered, washed with excess water, and finally dried at 50 °C under vacuum. The white solid obtained was denoted as POP-AO. POP-AO was treated with 3% (w/w) potassium hydroxide aqueous solution at room temperature for 36 h before adsorption tests. POP-$p$NH$_2$-AO and POP-$o$NH$_2$-AO were synthesized follow the same procedures except that POP-$p$NH$_2$-CN or POP-$o$NH$_2$-CN was used instead of POP-CN.

**Data availability**. The authors declare that all the data supporting the findings of this study are available within the article (and Supplementary Information files), or available from the corresponding author on reasonable request. The X-ray crystallographic coordinates for structures reported in this study have been deposited at the Cambridge Crystallographic Data Center (CCDC), under deposition numbers 1547954. These data can be obtained free of charge from the Cambridge Crystallographic Data Center via http://www.ccdc.cam.ac.uk/data_request/cif.

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

# ARTICLE

18. Wang, L. L. et al. Ultrafast high-performance extraction of uranium from seawater without pretreatment using an acylamide- and carboxyl-functionalized metal-organic framework. *J. Mater. Chem. A* **3**, 13724–13730 (2015).

19. Zhou, L. et al. A protein engineered to bind uranyl selectively and with femtomolar affinity. *Nat. Chem.* **6**, 236–241 (2014).

20. Odoh, S. O. et al. $UO_2^{2+}$ uptake by proteins: understanding the binding features of the super uranyl binding protein and design of a protein with higher affinity. *J. Am. Chem. Soc.* **136**, 17484–17497 (2014).

21. Vukovic, S. & Hay, B. P. De novo structure-based design of bis-amidoxime uranophiles. *Inorg. Chem.* **52**, 7805–7810 (2013).

22. Franczyk, T. S., Czerwinski, K. R. & Raymond, K. N. Stereognostic coordination chemistry. 1. The design and synthesis of chelators for the uranyl ion. *J. Am. Chem. Soc.* **114**, 8138–8146 (1992).

23. Alexandratos, S. D., Zhu, X., Florent, M. & Sellin, R. Polymer-supported bifunctional amidoximes for the sorption of uranium from seawater. *Ind. Eng. Chem. Res.* **55**, 4208–4216 (2016).

24. Li, R. et al. Optimization of molar content of amidoxime and acrylic acid in UHMWPE fibers for improvement of seawater uranium adsorption capacity. *J. Radioanal. Nucl. Chem.* **311**, 1771–1779 (2017).

25. Slater, A. G. & Cooper, A. I. Function-led design of new porous materials. *Science* **348**, aaa988 (2015).

26. Das, S., Heasman, P., Ben, T. & Qiu, S. Porous organic materials: strategic design and structure-function correlation. *Chem. Rev.* **117**, 1515–1563 (2017).

27. Xu, Y., Jin, S., Xu, H., Nagai, A. & Jiang, D. Conjugated microporous polymers: design, synthesis and application. *Chem. Soc. Rev.* **42**, 8012–8031 (2013).

28. Wu, D. et al. Design and preparation of porous polymers. *Chem. Rev.* **112**, 3959–4015 (2012).

29. Li, B., Zhang, Y., Ma, D., Shi, Z. & Ma, S. Mercury nano-trap for effective and efficient removal of mercury (II) from aqueous solution. *Nat. Commun.* **5**, 5537 (2014).

30. Byun, J., Patel, H. A., Thirion, D. & Yavuz, C. T. Charge-specific size-dependent separation of water-soluble organic molecules by fluorinated nanoporous networks. *Nat. Commun.* **7**, 13377 (2016).

31. Astheimer, L., Schenk, H. J., Witte, E. G. & Schwochau, K. Development of sorbers for the recovery of uranium from seawater. Part 2. The accumulation of uranium from seawater by resins containing amidoxime and imidoxime functional groups. *Sep. Sci. Technol.* **18**, 307–339 (1983).

32. Kelley, S. P., Barber, P. S., Mullins, P. H. K. & Rogers, R. D. Structural clues to $UO_2^{2+}/VO_2^{+}$ competition in seawater extraction using amidoxime-based extractants. *Chem. Commun.* **50**, 12504–12507 (2014).

33. Eloy, F. & Lenaers, R. The chemistry of amidoximes and related compounds. *Chem. Rev.* **62**, 155–183 (1962).

34. Bai, Z.-Q. et al. Introduction of amino groups into acid-resistant MOFs for enhanced U(VI) sorption. *J. Mater. Chem. A* **3**, 525–534 (2015).

35. Doidge, E. D. et al. A simple primary amide for the selective recovery of gold from secondary resources. *Angew. Chem. Int. Ed.* **55**, 12436–12439 (2016).

36. Sun, Q. et al. Highly efficient heterogeneous hydroformylation over Rh-metalated porous organic polymers: synergistic effect of high ligand concentration and flexible framework. *J. Am. Chem. Soc.* **137**, 5205–5209 (2015).

37. Sun, Q. et al. Superhydrophobicity: constructing homogeneous catalysts into superhydrophobic porous frameworks to protect them from hydrolytic degradation. *Chem* **1**, 628–639 (2016).

38. Das, S. et al. Novel poly(imide dioxime) sorbents: development and testing for enhanced extraction of uranium from natural seawater. *Chem. Eng. J.* **298**, 125–135 (2016).

39. Sun, Q., Dai, Z., Meng, X. & Xiao, F.-S. Porous polymer catalysts with hierarchical structures. *Chem. Soc. Rev.* **44**, 6018–6034 (2015).

40. Sun, M.-H. et al. Applications of hierarchically structured porous materials from energy storage and conversion, catalysis, photocatalysis, adsorption, separation, and sensing to biomedicine. *Chem. Soc. Rev.* **45**, 3749–3563 (2016).

41. Pan, H.-B. et al. Towards understanding KOH conditioning of amidoxime-based polymer adsorbents for sequestering uranium from seawater. *RSC Adv.* **5**, 100715–100721 (2015).

42. Silver, M. A. et al. Why is uranyl formohydroxamate red. *Inorg. Chem.* **54**, 5280–5284 (2015).

43. Barber, P. S., Kelley, S. P., Griggs, C. S., Wallace, S. & Rogers, R. D. Surface modification of ionic liquid-spun chitin fibers for the extraction of uranium from seawater: seeking the strength of chitin and the chemical functionality of chitosan. *Green Chem.* **16**, 1828–1836 (2014).

44. Rao, L. *Recent International R&D Activities in the Extraction of Uranium from Seawater Report LBNL-4034E* (Lawrence Berkeley National Laboratory, Berkeley, CA, 2010).

45. Vukovic, S., Watson, L. A., Kang, S. O., Custelcean, R. & Hay, B. P. How amidoximate binds the uranyl cation. *Inorg. Chem.* **51**, 3855–3859 (2012).

46. Wegner, S. V., Boyaci, H., Chen, H., Jensen, M. P. & He, C. Engineering a uranyl-specific binding protein from NikR. *Angew. Chem. Int. Ed.* **48**, 2339–2341 (2009).

47. Mehio, N. et al. Acidity of the amidoxime functional group in aqueous solution: a combined experimental and computational study. *J. Phys. Chem. B* **119**, 3567–3576 (2015).

48. Mehio, N. et al. Quantifying the binding strength of salicylaldoxime-uranyl complexes relative to competing salicylaldoxime-transition metal ion complexes in aqueous solution: a combined experimental and computational study. *Dalton. Trans.* **45**, 9051–9064 (2016).

49. Wang, C. Z. et al. Theoretical insights on the interaction of uranium with amidoxime and carboxyl groups. *Inorg. Chem.* **53**, 9466–9476 (2014).

50. Ramamoorthy, S. & Santappa, M. Stability constants of some uranyl complexes. *Bull. Chem. Soc. Jpn.* **41**, 1330–1333 (1968).

51. Martell, A. E. & Smith, R. M. *Critical Stability Constant Database, 46* (National Institute of Science and Technology (NIST), Gaithersburg, MD, 2003).

52. Manos, M. J. & Kanatzidis, M. G. Layered metal sulfides capture uranium from seawater. *J. Am. Chem. Soc.* **134**, 16441–16446 (2012).

53. Barber, P. S., Kelley, S. P. & Rogers, R. D. Highly selective extraction of the uranyl ion with hydrophobic amidoxime functionalized ionic liquids via $\eta^2$ coordination. *RSC Adv.* **2**, 8526–8530 (2012).

54. Zhang, A., Asakura, T. & Uchiyama, G. The adsorption mechanism of uranium(VI) from seawater on a macroporous fibrous polymeric adsorbent containing amidoxime chelating functional group. *React. Funct. Polym.* **57**, 67–76 (2003).

## Acknowledgements

This work was supported by the DOE Office of Nuclear Energy's Nuclear Energy University Program (Grant No. DE-NE0008281). Work by L.D.E and C.W.A was supported financially by the Division of Chemical Sciences, Geosciences, and Biosciences, Office of Basic Energy Sciences, U.S. Department of Energy. This manuscript has been authored by UT-Battelle, LLC under Contract No. DE-AC05-00OR22725 with the U.S. Department of Energy. The United States Government retains and the publisher, by accepting the article for publication, acknowledges that the United States Government retains a non-exclusive, paid-up, irrevocable, worldwide license to publish or reproduce the published form of this manuscript, or allow others to do so, for United States Government purposes. The Department of Energy will provide public access to these results of federally sponsored research in accordance with the DOE Public Access Plan (http://energy.gov/downloads/doe-public-access-plan). Use of the Stanford Synchrotron Radiation Lightsource, SLAC National Accelerator Laboratory, is supported by the U.S. Department of Energy, Office of Science, Office of Basic Energy Sciences under Contract No. DE-AC02-76SF00515.

## Author contributions

Q.S. and S.M conceived and designed the research. Q.S. performed the synthesis. Q.S. and B.A. carried out the adsorption tests. A.I. and V.B. contributed to DFT calculation. L. D.E and C.W.A performed and analyzed the EXASF. L.W. contributed to the analysis of single crystals. All authors participated in drafting the paper, and gave approval to the final version of the manuscript.

## Additional information

**Competing interests:** The authors declare no competing interests.

