## [Peer Review File · Nature Communications]

Reviewers' comments:

Reviewer #1 (Remarks to the Author):

In this work, the authors prepared a kind of amino group modified POP, POP-oNH₂-AO, for removal of uranium from water samples. The as-prepared material was well characterized and showed high adsorption capacity towards uranium. However, "Bio-inspired" indicated in the title was not discussed extensively in the main text. In addition, the interferences from other heavy metals such as Cu, Zn, Fe, etc. which also has potential interaction with amino group were not studied. Overall, it is a complete manuscript, and the material has the potential applicability in uranium removal, but it is more suitable for submission to a specific journal.

Specific comments:

1. Since the authors claimed that "the resultant adsorbent constructed with an amino group in the ortho position relative to amidoxime displayed extraordinary affinity for uranyl, making it one of the best uranium adsorbents reported thus far", the characterization of POP-oNH₂-AO should be moved into the manuscript and the characterization of POP-AO should be moved into supporting information.
2. The concentration of uranium in nuclear waste water and environment water (seawater) should be discussed in the text.
3. The effect of pH as well as the effect of the ration of solution volume (V) to adsorbent mass (m) on the removal efficiency of uranium should be investigated considering that the change of pH will lead to the change of uranium species (as shown in Table S5).
4. What about the selectivity of the prepared materials? Could the removal efficiency of the uranium with this material be influenced by other ions in the sea water, such as Cu, Zn, Fe, Pu, Ra, Po, Pb, Cs and Sr?
5. On page 11, For the investigation of the uranium capture from seawater, the uranium should be spiked into real seawater other than simulate seawater.
6. The description on adsorption ability of adsorbent in seawater on page 19 should be moved to the section of "Uranium capture from simulated seawater".
7. The authors should refine and readjust the structure of the manuscript to make it more reasonable.
8. What was the advantages of the prepared materials for the removal of uranium compared with other reported materials? This issue should be stated more clearly to show the advantages of the proposed method.
9. In the section of "Discussion", the authors mentioned that after 56 days of seawater exposure, the amount of uranium enriched in the adsorbent was determined by ICP-MS analysis after being digested by aqua regia, confirming the superior performance of POP-oNH₂-AO and thereby showing great promise for practical applications. Why did the authors choose the exposure time of 56 days to investigate the amount of uranium enriched in the adsorbent?

Reviewer #2 (Remarks to the Author):

Highly efficient extraction of uranium from aqueous solution or seawater is of significance for producing nuclear fuel as well as minimizing the adverse impacts of uranium on the environment and human health. This manuscript reported an effective approach by constructing amidoxime and amino groups as chelating moieties into porous organic polymers (POP) to enhance the affinity to uranyl. The developed POPs were sufficiently characterized. And the adsorption behaviors of uranium on the materials were properly studied in both aqueous solution and simulated/real seawater. The results are interesting, showing a promising of such materials for further practical applications. Therefore, I would suggest the acceptance after minor revisions addressing the following concerns:

1. As shown in Table 1, the BET surface areas of as-synthesized POPs decreased dramatically from 834 m² g⁻¹ (POP-CN) to 415 m² g⁻¹ (POP-oNH₂-AO). Even so, the uranyl uptake capacity was

markedly improved by introducing these -AO and -NH₂ groups. Considering that the loss of pore volumes seems to contribute an adverse effect to uranium sorption, please make clearer the influence of the porosity change in these POPs framework on the performance of uranyl extraction.

2. As depicted in Fig. 2, the adsorption capacities of 580 mg g⁻¹ for POP-pNH₂-AO is higher than 530 mg g⁻¹ for POP-oNH₂-AO in aqueous solution. But the POP-oNH₂-AO was superior than POP-pNH₂-AO with regard to both kinetics and removal efficiency. In addition, the experimental or DFT calculation results also confirmed the higher affinity of amino group in ortho position than para position. Please add more discussion about this discrepancy.

3. The authors describe the concept about bio-inspired creation of uranium "nano-traps". The amidoxime/aminio groups were chosen as chelating sites to uranyl ions. Generally, many natural or artificially designed protein sequences for specific metal ion binding frequently involved the carboxylate donors, for instance, aspartates and glutamates (such as super uranyl-binding protein, Zhou L., et al. *Nat. Chem.* 6, 236-241(2014)). Meanwhile, the introduced carboxyl units in porous MOF materials have notably improved the sorption capacity and selective extraction of uranium in aqueous solution (such as carboxyl functionalized MOFs, Wang L., et al. *J Mater. Chem. A* 3, 13724-13730 (2015), and Li L., et al. *ACS Appl. Mater. Interfaces* 8, 31032-31041(2016)). It would be helpful to the readers if more discussion or comments are added about these two different strategies, in which carboxyl and amidoxime/aminio groups were introduced.

4. As the desired property, the selectivity over other metal ions is important for the practical applications of such materials. How about the selectivity of these POPs toward uranium over other competing metal ions, such as transition heavy metal ions (eg. Cu²⁺, Fe³⁺, Co²⁺, Pb²⁺) and lanthanide (La³⁺, Ce³⁺, Sm³⁺, etc) or actinide elements? The competitive sorption experiments with the existence of above interfering ions would help to reveal the intrinsic selectivity on the uranyl ion.

5. The developed POPs were applied in the simulated/real seawater containing excess Na⁺. Do the Mg²⁺ and Ca²⁺, which are also abundant in the ocean, have an obvious influence on the sorption process?

6. How is the chemical stability of the absorbents under intense acidic or basic pH environments that were frequently used in many elution process like 1 M HNO₃(aq), NaCO₃(aq), or NaOH(aq) solution. For recycling and reusing materials, is there any suitable eluent for the uranium desorption?

Response to Reviewers' comments:

We greatly appreciate the constructive comments and suggestions from both reviewers, and we have revised the manuscript accordingly as detailed in the responses below. The corresponding changes have been highlighted in yellow in the main text and supplementary information.

Reviewer #1:

Comment 1: In this work, the authors prepared a kind of amino group modified POP, POP-oNH₂-AO, for removal of uranium from water samples. The as-prepared material was well characterized and showed high adsorption capacity towards uranium. However, “Bio-inspired” indicated in the title was not discussed extensively in the main text. In addition, the interferences from other heavy metals such as Cu, Zn, Fe, etc. which also has potential interaction with amino group were not studied. Overall, it is a complete manuscript, and the material has the potential applicability in uranium removal, but it is more suitable for submission to a specific journal.

Response: We thank the reviewer for taking the time to review our manuscript and valuable comments. Per the reviewer’s suggestion, we have emphasized more about the “bio-inspired” design concept in the main text. The studies on the interferences from other heavy metals have also been conducted, which together with other comments from the reviewer have been responded point-by-point detailed as follows.

We appreciate the positive comments “it is a complete manuscript, and the material has the potential applicability in uranium removal”, and we believe Nature Communications is the suitable home for our work.

Comment 2: Since the authors claimed that “the resultant adsorbent constructed with an amino group in the ortho position relative to amidoxime displayed extraordinary affinity for uranyl, making it one of the best uranium adsorbents reported thus far”, the characterization of POP-oNH₂-AO should be moved into the manuscript and the characterization of POP-AO should be moved into supporting information.

Response: We thank the reviewer for the valuable suggestion. We have moved the characterization of POP-oNH₂-AO into the main text and the details of POP-AO into supplementary information.

Comment 3: The concentration of uranium in nuclear waste water and environment water (seawater) should be discussed in the text.

Response: We thank the reviewer for the comment. The concentration of uranium in nuclear waste water and environment water (seawater) have been included in the text.

Comment 4: The effect of pH as well as the effect of the ratio of solution volume (V) to adsorbent mass (m) on the removal efficiency of uranium should be investigated considering that the change of pH will lead to the change of uranium species (as shown in Table S5).

Response: We appreciate the reviewer for the valuable comments. Per the reviewer's suggestion, the effect of pH and the ratio of solution volume (V) to adsorbent mass on the uranium removal efficiency have been carefully investigated, as exemplified by POP-oNH₂-AO. It is shown that uranium adsorption is strongly dependent on the pH value of the solution. Sorption of uranium by POP-oNH₂-AO in distilled water was measured as a function of pH over the range of 2 to 9, with the maximum observed around a pH of 6.

Moreover, to evaluate the uranium removal efficiency of POP-oNH₂-AO, the effect of the uranium contaminated potable water volume (V, C₀ = 1000 ppb) to adsorbent mass (m) used was carefully investigated. It is shown that POP-oNH₂-AO exhibits excellent affinity towards uranium and more than 99.9% of uranium species can be extracted by a single treatment at a very high V/m ratio of 500000 mL g⁻¹ (the residual uranium concentration is less than 1 ppb).

Comment 5: What about the selectivity of the prepared materials? Could the removal efficiency of the uranium with this material be influenced by other ions in the sea water, such as Cu, Zn, Fe, Pu, Ra, Po, Pb, Cs and Sr?

Response: We are thankful to the reviewer for the comments. The removal efficiency of the synthesized materials towards uranium species in the presence of various interfering ions, such as transition heavy metal ions (eg. Cu²⁺, Fe³⁺, Co²⁺, Pb²⁺, Zn²⁺), lanthanide (La³⁺, Ce³⁺, Sm³⁺, etc), as well as Mg²⁺ and Ca²⁺ has been investigated. The adsorption tests were performed using a potable water sample containing uranium and the ions aforementioned with nearly equal concentrations (ca. 1000 ppb). It is shown that after a single treatment, uranium was removed with over 99% efficiency by POP-oNH₂-AO when added at a phase ratio of 40000 mL g⁻¹ over 1 h contact time.

Comment 6: On page 11, for the investigation of the uranium capture from seawater, the uranium should be spiked into real seawater other than simulate seawater.

Response: We thank the reviewer for the valuable comment. Per the reviewer's suggestion, we examined their performance in seawater samples spiked with 10.3 ppm uranium to take the impact of other competing ions on the uranium capture into account. Reduced uptake capacities from 20-30% for POP-AO and POP-pNH₂-AO were detected relative to the uptake values of uranium spiked simulated seawater, giving 143 mg and 202 mg uranium per gram of adsorbent, respectively. However, less than 5% decrease in uptake capacity was observed for POP-oNH₂-AO (290 mg g⁻¹ vs 276 mg g⁻¹), indicative of its excellent affinity towards uranium.

Comment 7: The description on adsorption ability of adsorbent in seawater on page 19 should be moved to the section of "Uranium capture from simulated seawater".

Response: We appreciate the reviewer for the valuable suggestion. We have moved the description on the adsorbents performance of seawater uranium mining to the section of "Uranium capture from simulated seawater".

Comment 8: The authors should refine and readjust the structure of the manuscript to make it more reasonable.

Response: We thank the reviewer for the comment. Per the reviewer's suggestion, we have refined the structure of our manuscript.

Comment 9: What was the advantages of the prepared materials for the removal of uranium compared with other reported materials? This issue should be stated more clearly to show the advantages of the proposed method.

Response: We are thankful for the reviewer's valuable comment. Per the reviewer's suggestion, the merits of our materials in relation to the reported ones have been emphasized more.

Comment 10: In the section of "Discussion", the authors mentioned that after 56 days of seawater exposure, the amount of uranium enriched in the adsorbent was determined by ICP-MS analysis after being digested by aqua regia, confirming the superior performance of POP-*o*NH₂-AO and thereby showing great promise for practical applications. Why did the authors choose the exposure time of 56 days to investigate the amount of uranium enriched in the adsorbent?

Response: We thank the reviewer for the comment. To ensure valid comparisons between different adsorbents in the enrichment of uranium form seawater, a period of 56 days exposure is set up by the U. S. Department of Energy.

Reviewer #2:

Comment 1: Highly efficient extraction of uranium from aqueous solution or seawater is of significance for producing nuclear fuel as well as minimizing the adverse impacts of uranium on the environment and human health. This manuscript reported an effective approach by constructing amidoxime and amino groups as chelating moieties into porous organic polymers (POP) to enhance the affinity to uranyl. The developed POPs were sufficiently characterized. And the adsorption behaviors of uranium on the materials were properly studied in both aqueous solution and simulated/real seawater. The results are interesting, showing a promising of such materials for further practical applications. Therefore, I would suggest the acceptance after minor revisions addressing the following concerns:

Response: We appreciate the reviewer's high comments and support of our work.

Comment 2: As shown in Table 1, the BET surface areas of as-synthesized POPs decreased dramatically from 834 m² g⁻¹ (POP-CN) to 415 m² g⁻¹ (POP-*o*NH₂-AO). Even so, the uranyl uptake capacity was markedly improved by introducing these -AO and -NH₂ groups. Considering that the loss of pore volumes seems to contribute an adverse effect to uranium sorption, please make clearer the influence of the porosity change in these POPs framework on the performance of uranyl extraction.

Response: We are thankful to the reviewer for the criticism. We agree with the reviewer that the surface area of materials has impact on their sorption performance towards the guest species, in particular for physical adsorption. However, in chemical adsorption, the binding affinity of the chelating group towards the guest species may play a more important role on the adsorbent overall performance in terms of uptake capacity and selectivity. Considering that POP-AO with higher surface area (696 m² g⁻¹) and pore volume (0.52 cm³ g⁻¹) in comparison with other materials tested, POP-*p*NH₂-AO (397 m² g⁻¹, 0.22 cm³ g⁻¹) and POP-*o*NH₂-AO (415 m² g⁻¹, 0.22 cm³ g⁻¹), does not show high adsorption performance, we thereby reasoned that rather than the textural features, the different coordination environments of these materials seem to

be responsible for the observed distinct disparity in uranium capture. We have emphasized more in the revised manuscript regarding this point.

Comment 3: As depicted in Fig. 2, the adsorption capacities of 580 mg g^{-1} for POP-*p*NH₂-AO is higher than 530 mg g^{-1} for POP-*o*NH₂-AO in aqueous solution. But the POP-*o*NH₂-AO was superior than POP-*p*NH₂-AO with regard to both kinetics and removal efficiency. In addition, the experimental or DFT calculation results also confirmed the higher affinity of amino group in ortho position than para position. Please add more discussion about this discrepancy.

Response: We appreciate the reviewer for the valuable comment. The highest uranyl capture capacity from water given by POP-*p*NH₂-AO among the tested adsorbents can be reasonably attributed to the separate coordination between the uranyl-amidoxime and uranyl-amine in POP-*p*NH₂-AO. Interaction between the amino group and uranium is expected due to the successful uranyl extraction solely on an amine-based MOF (see reference 34). In the case of POP-*o*NH₂-AO, the amino group participates in the complex formation, serving as a reinforce group to enhance the coordinative interaction between amidoxime and uranyl, which does not bind with uranyl proven by the single crystal structure. However, in the presence of other competing ions, such as simulated seawater and seawater, sorbent material (POP-*o*NH₂-AO) with higher binding affinity towards uranium shows superior performance since the relatively weak binding sites (amino group) may be unable to capture the target ions. We have included the corresponding discussion in the revised manuscript.

Comment 4: The authors describe the concept about bio-inspired creation of uranium “nano-traps”. The amidoxime/aminio groups were chosen as chelating sites to uranyl ions. Generally, many natural or artificially designed protein sequences for specific metal ion binding frequently involved the carboxylate donors, for instance, aspartates and glutamates (such as super uranyl-binding protein, Zhou L., et al. Nat. Chem. 6, 236-241(2014)). Meanwhile, the introduced carboxyl units in porous MOF materials have notably improved the sorption capacity and selective extraction of uranium in aqueous solution (such as carboxyl functionalized MOFs, Wang L., et al. J Mater. Chem. A 3, 13724-13730 (2015), and Li L., et al. ACS Appl. Mater. Interfaces 8, 31032-31041(2016)). It would be helpful to the readers if more discussion or comments are added about these two different strategies, in which carboxyl and amidoxime/aminio groups were introduced.

Response: We are thankful for the reviewer’s suggestion. We have included the discussion about the developed strategies to improve the performance of adsorbents in uranium capture and the corresponding references have been properly cited.

Comment 5: As the desired property, the selectivity over other metal ions is important for the practical applications of such materials. How about the selectivity of these POPs toward uranium over other competing metal ions, such as transition heavy metal ions (eg. Cu²⁺, Fe³⁺, Co²⁺, Pb²⁺) and lanthanide (La³⁺, Ce³⁺, Sm³⁺, etc) or actinide elements? The competitive sorption experiments with the existence of above interfering ions would help to reveal the intrinsic selectivity on the uranyl ion.

Response: We appreciate the reviewer for the constructive comment. Per the reviewer’s suggestion, the competitive sorption experiments in the presence of various ions have been studied. The removal efficiency of the synthesized materials towards uranium species in the presence of interfering ions, such as transition heavy metal ions (eg. Cu²⁺, Fe³⁺, Co²⁺, Pb²⁺,

Zn²⁺), lanthanide (La³⁺, Ce³⁺, Sm³⁺, etc), as well as Mg²⁺ and Ca²⁺ has been investigated. The adsorption tests were performed using a potable water sample containing uranium and the ions aforementioned with nearly equal concentrations (1000 ppb). It is shown that after a single treatment, uranium was removed with over 99% efficiency by POP-oNH₂-AO when added at a phase ratio of 40000 mL g⁻¹ over 1 h contact time

Comment 6: The developed POPs were applied in the simulated/real seawater containing excess Na⁺. Do the Mg²⁺ and Ca²⁺, which are also abundant in the ocean, have an obvious influence on the sorption process?

Response: We thank the reviewer for the comment. The influence of Mg²⁺ and Ca²⁺ on the uranium removal efficiency from aqueous solutions has been studied (see last comment). To take the impact of Mg²⁺, Ca²⁺, and other ions on the uranium adsorption into account, we compared the adsorbents uptake capacities from uranium spiked simulated seawater and real seawater. Reduced uptake capacities from 20-30% for POP-AO and POP-pNH₂-AO were detected relative to the uptake values of uranium spiked simulated seawater, giving 143 mg and 202 mg uranium per gram of adsorbent, respectively. However, less than 5% decrease in uptake capacity was observed for POP-oNH₂-AO (290 mg g⁻¹ vs 276 mg g⁻¹), indicative of its excellent affinity towards uranium.

Comment 7: How is the chemical stability of the adsorbents under intense acidic or basic pH environments that were frequently used in many elution process like 1 M HNO₃(aq), NaCO₃(aq), or NaOH(aq) solution. For recycling and reusing materials, is there any suitable eluent for the uranium desorption?

Response: We appreciate the reviewer for the comment. The chemical stability of the adsorbents against intense acidic or basic aqueous solutions ranging from 1 M HNO₃ and 1 M NaOH as well as in 1 M Na₂CO₃ was evaluated, as exemplified by POP-oNH₂-AO. The adsorbent is stable under the above conditions as evidenced by the fact that negligible change is observed in the IR spectra of the material before and after being soaked under those solutions for 24 h.

Multi-repeated experiments, as demonstrated by POP-oNH₂-AO, indicate that the laden uranium species on the adsorbent could be easily eluted with 1 M Na₂CO₃ and the material can be recycled up to two-times with negligible loss of performance.

Again we thank the editor and reviewers for the constructive comments and suggestions, which have made our manuscript much improved.

Reviewers' comments:

Reviewer #1 (Remarks to the Author):

The authors replied almost the comments raised by the reviewers and revised the manuscript accordingly, I have only comment for this paper before the consideration of its publication.

1. The authors evaluated the ability of POP-oNH₂-AO to eliminate uranium in the presence of various interfering ions (Cu²⁺, Fe³⁺, Co²⁺, Pb²⁺, Zn²⁺, La³⁺, Ce³⁺, Sm³⁺, Mg²⁺ and Ca²⁺) using a potable water sample containing uranium and the ions aforementioned with nearly equal concentrations (1000 ppb). However, as the authors claimed in the text, "uranium is arduous to selectively capture due to its extremely low concentration", and the concentration of above interfering ions in seawater would be much higher than uranium, so a potable water sample containing uranium and the ions with equal concentrations is not a suitable sample to evaluate the selectivity of the prepared material. In addition, in the nuclear waste, there are other elements such as Pu, Ra, Po, Pb, Cs and Sr, the effect of these interferences need to be investigated.

Response to Reviewers' comments:

We greatly appreciate the positive comments and constructive suggestions from the reviewer, and we have revised the manuscript accordingly as detailed in the responses below. The corresponding changes have been highlighted in yellow in the main text.

Reviewer #1:

Comment 1: The authors replied almost the comments raised by the reviewers and revised the manuscript accordingly, I have only comment for this paper before the consideration of its publication.

Response: We appreciate the reviewer for taking the time to review our manuscript again and positive comments. The concerns raised by the reviewer have been responded point-by-point detailed as follows.

Comment 2: 1. The authors evaluated the ability of POP-oNH₂-AO to eliminate uranium in the presence of various interfering ions (Cu²⁺, Fe³⁺, Co²⁺, Pb²⁺, Zn²⁺, La³⁺, Ce³⁺, Sm³⁺, Mg²⁺, and Ca²⁺) using a potable water sample containing uranium and the ions aforementioned with nearly equal concentrations (1000 ppb). However, as the authors claimed in the text, "uranium is arduous to selectively capture due to its extremely low concentration", and the concentration of above interfering ions in seawater would be much higher than uranium, so a potable water sample containing uranium and the ions with equal concentrations is not a suitable sample to evaluate the selectivity of the prepared material.

Response: We thank the reviewer for the constructive comments. Per the reviewer's suggestion, the removal efficiency of POP-oNH₂-AO towards uranium species in the presence of a large excess of interfering ions, such as transition heavy metal ions (Cu²⁺, Fe³⁺, Co²⁺, Pb²⁺, Zn²⁺), lanthanides (La³⁺, Ce³⁺, Sm³⁺), radioactive ions (Cs⁺, Sr²⁺), as well as Mg²⁺, Ca²⁺, has been investigated. The adsorption tests were performed using a potable water sample containing uranium (ca. 1 ppm) and the ions aforementioned (ca. 500 ppm). It is shown that after a single treatment, uranium was removed with over 99% efficiency by POP-oNH₂-AO when added at a phase ratio of 10000 mL g⁻¹ over 1 h contact time, indicative its excellent selectivity towards uranium.

Comment 3: In addition, in the nuclear waste, there are other elements such as Pu, Ra, Po, Pb, Cs and Sr, the effect of these interferences need to be investigated.

Response: We appreciate the reviewer for the valuable comments. Given the strict regulations, we do not have access to Pu, Ra, and Po. Nonetheless, we have evaluated the uranium removal efficiency in the presence of a large excess of other radioactive ions Pb²⁺, Cs⁺, and Sr²⁺, as

detailed in the comment 2. The results indicate that these interferences have negligible impact on the removal efficiency of POP-oNH₂-AO towards uranium.

Again we thank the reviewer for the constructive suggestions, which have made our manuscript further improved.

REVIEWERS' COMMENTS:

Reviewer #1 (Remarks to the Author):

The authors replied the comments raised by the reviewer and revised the manuscript accordingly, therefore, I think this paper can be published in the present form.

Response to Reviewers' comments:

Reviewer #1

Comment 1: The authors replied the comments raised by the reviewer and revised the manuscript accordingly, therefore, I think this paper can be published in the present form.

Response: We are grateful to the reviewer for taking time to evaluate our work and support from the reviewer.